**∂ | Open Peer Review** | *Clinical Microbiology | Observation*

# Real-world evaluation of the Lucira Check-It COVID-19 loop-mediated amplification (LAMP) test

Elizabeth Simms,[1,2,3] Gregory R. McCracken,[3] Todd F. Hatchette,[1,2,3] Shelly A. McNeil,[1] Ian Davis,[1,2,3] Noella Whelan,[4] Angela Keenan,[5] Jason J. LeBlanc,[1,2,3] Glenn Patriquin[1,2,3]

**ABSTRACT** In hospitals during the COVID-19 pandemic, laboratory testing was important to reduce SARS-CoV-2 transmissions, particularly for high-risk settings like the emergency department and pre-operative settings and for the safe return to work of exposed healthcare workers (HCWs). For these applications, delayed test results from laboratory nucleic acid amplification tests (NAATs) posed a barrier to maximizing efficient patient flow and minimizing staffing shortages. This quality improvement project sought to evaluate the performance of the Lucira Check-It COVID-19 Test, a rapid diagnostic test that used NAAT technology (NAAT-RDT). Using 10-fold serial dilutions of SARS-CoV-2, the analytical sensitivity of the NAAT-RDT was assessed against standard NAATs used for routine diagnostic testing. Clinical performance was assessed at two Nova Scotia hospitals in 405 cases with paired swabs tested by NAAT-RDT and laboratory-based NAATs. These represented three distinct populations: patients presenting to the emergency department ($n = 208$), patients in the pre-operative setting ($n = 158$), and patients presenting to community testing sites ($n = 38$). The analytical sensitivity of the NAAT-RDT and other laboratory NAATs was comparable. During clinical evaluation, the overall sensitivity and specificity were 92.9% and 98.3%, respectively, with little variation between settings. The Lucira NAAT-RDT is a portable and self-contained device that provides an easily interpreted result within 30 minutes following a bilateral nasal swab collection. Its performance was shown to be acceptable for use in three settings in this quality improvement project, facilitating patient flow and management.

**IMPORTANCE** In hospitals during the COVID-19 pandemic, laboratory testing was important to reduce SARS-CoV-2 transmissions, while facilitating patient flow in the emergency department and pre-operative settings, and allowing for the safe return to work of exposed healthcare workers. Delayed test results from laboratory nucleic acid amplification tests (NAATs) posed a barrier to maximizing efficient patient flow and minimizing staffing shortages. This quality improvement project sought to evaluate the analytical and clinical performance of the Lucira Check-It COVID-19 Test, a point-of-care test that used NAAT technology, in the perioperative setting, emergency department, and community testing sites. We found the Lucira Check-It to have comparable performance to laboratory NAATs. It can be employed with little training for specimen collection, processing, and interpretation, and at a cost justifiable from the resources saved from avoiding sample transport and laboratory testing.

**KEYWORDS** COVID-19, SARS-CoV-2, diagnostic, loop-mediated amplification (LAMP), point-of-care (POC)

Address correspondence to Elizabeth Simms, elizabeth.simms@nshealth.ca, or Glenn Patriquin, glenn.patriquin@nshealth.ca.

The authors declare no conflict of interest.

The COVID-19 pandemic disrupted the organization of patient care delivery in healthcare settings (1, 2). In hospitals, SARS-CoV-2 testing was implemented to reduce transmission; however, delays in laboratory test results impacted patient flow

in areas like the emergency department (ED) or pre-operative settings (1, 2). Additional strains on healthcare services arose from healthcare worker absenteeism due to SARS-CoV-2 exposure (2). To address these challenges, various rapid diagnostic tests (RDTs) were developed for SARS-CoV-2 detection with point-of-care (POC) applications (3–6). Antigen-based RDTs (Ag-RDTs) often produce results within 15 minutes, but their sensitivity is reduced compared to laboratory nucleic acid amplification tests (NAATs), and repeat testing over time is required to achieve maximal sensitivity (6). NAAT-based RDTs (NAAT-RDTs) were also developed that provide rapid molecular detection (<1 h) of SARS-CoV-2 with increased sensitivity compared to Ag-RDTs but are often limited by their scalability (3–6). The Lucira Check-It COVID-19 Test (Lucira Health, Inc., Emery-ville, CA) is a NAAT-RDT based on the reverse transcription loop-mediated amplification (RT-LAMP) technology targeting the SARS-CoV-2 N gene and a region spanning the open reading frames (Orf) 7b/8 (3, 4). The amplification of these targets if present leads to a pH change detected by colorimetric detection (3, 4). Like other isothermal NAAT-RDT technologies (3), this easy-to-use portable device provides interpreted results within 30 minutes of swab collection (3, 4). These features make it an appealing alternative to laboratory NAATs for SARS-CoV-2 testing where rapid results can facilitate patient flow or occupational health decision-making (7–11). Lucira was the first commercial NAAT-RDT available in Canada, but little data existed on its performance (4).

## Objectives

This quality improvement project sought to determine the analytical and clinical performance of the Lucira NAAT-RDT in the pre-operative setting, ED, and community testing sites.

## Study design

The analytical sensitivity (Table 1) of the Lucira NAAT-RDT device was determined by performing replicate ($n = 5$) 10-fold serial dilutions in universal transport media (UTM; Copan Diagnostics Inc., Murrieta, CA) of a SARS-CoV-2-positive oropharyngeal/bilateral nares swab (12). The NAAT-RDT nasal swab was placed into the virus dilutions for 10 seconds, and the remaining processing steps were performed according to the manufacturer's instructions. These dilutions were also used to compare the analytical sensitivity of the NAAT-RDT to another isothermal NAAT used in Canada [the ID NOW (Abbott Diagnostics, Scarborough ME)], as well as a laboratory-developed test (LDT) and

**TABLE 1** Analytical sensitivity of Lucira Check-It COVID-19 test compared to laboratory-based nucleic acid amplification tests[b]

| SARS-CoV-2 concentration (copies/mL)[a] | Proportion of positive results (avg. Ct value) | | | | | | |
| | | | | | | Cobas | |
| | LDT | ID NOW | Lucira | Aptima | Xpert | E | Orf1ab |
|---|---|---|---|---|---|---|---|
| 121,312 | 5/5 | 5/5 | 5/5 | 5/5 | 5/5 | 5/5 | 5/5 |
| | (28.9) | (N/A) | (N/A) | (N/A) | (23.7) | (27.9) | (28.2) |
| 6,578 | 5/5 | 5/5 | 5/5 | 5/5 | 5/5 | 5/5 | 5/5 |
| | (32.7) | (N/A) | (N/A) | (N/A) | (27.3) | (31.6) | (32.5) |
| 2,214 | 5/5 | 5/5 | 5/5 | 5/5 | 5/5 | 5/5 | 5/5 |
| | (35.6) | (N/A) | (N/A) | (N/A) | (30.4) | (33.0) | (34.3) |
| 917 | 2/5 | 2/5 | 4/5 | 5/5 | 5/5 | 5/5 | 5/5 |
| | (36.2) | (N/A) | (N/A) | (N/A) | (34.0) | (36.3) | (37.1) |
| 78 | 0/5 | 0/5 | 0/5 | 3/5 | 3/5 | 4/5 | 2/5 |
| | (N/A) | (N/A) | (N/A) | (N/A) | (37.4) | (38.1) | (39.2) |
| 8 | 0/5 | 0/5 | 0/5 | 0/5 | 0/5 | 0/5 | 0/5 |
| | (N/A) | (N/A) | (N/A) | (N/A) | (N/A) | (N/A) | (N/A) |
| 0 | 0/5 | 0/5 | 0/5 | 0/5 | 0/5 | 0/5 | 0/5 |
| | (N/A) | (N/A) | (N/A) | (N/A) | (N/A) | (N/A) | (N/A) |

[a]Quantified SARS-CoV-2 was provided by the National Microbiology Laboratory (NML), Winnipeg, MB.
[b] avg., average; Ct, cycle threshold; E, enveloped gene; LDT, laboratory-developed test; N/A, not applicable; Orf1ab, open reading frame 1ab.

the three standard NAAT assays used in clinical laboratories in Nova Scotia: (i) the Aptima SARS-CoV-2 Assay on the Panther System (Hologic Inc., San Diego, CA), (ii) the SARS-CoV-2 test on the Cobas 6800 instrument (Roche Diagnostics, Rotkreuz, Switzerland), and (iii) the Xpert Xpress SARS-CoV-2 test (Cepheid, Sunnyvale, CA) (5, 12–17).

The clinical performance of the NAAT-RDT assay was determined among three distinct populations. First, asymptomatic patients in the pre-operative setting with no known exposures to SARS-CoV-2 were tested on the day of the planned surgery. Second, ED patients whether asymptomatic or symptomatic were tested at the discretion of the attending team when an immediate result contributed to patient disposition and patient flow. Finally, testing was made at community testing sites for asymptomatic close contacts of COVID-19 cases at least 72 h prior and for return to work in a healthcare setting. Healthcare workers with close contact with positive SARS-CoV-2 cases were instructed to stay home for 7 days following exposure. Asymptomatic testing using the NAAT-RDT at 72 h allowed these individuals to return to work if test results were negative, reducing concerns of subsequent transmission and helping maintain hospital staffing. As a quality project with retrospective data review, the collection of paired specimens was not consistent between the NAAT-RDT and clinical laboratory-based NAATs. Comparator NAATs included one of three standard kits (Cobas 6800, Aptima, or Xpert) (12–15). Only individuals with a standard NAAT from a swab collected within 1 day of NAAT-RDT testing were included. Indeterminate NAAT results were scored based on the result of a follow-up test, and those without repeat testing were excluded. True positive and true negative results were defined by concordant results between the NAAT-RDT and one of the three standard NAATs. A negative NAAT-RDT test with a paired specimen testing positive using a standard NAAT was characterized as a false negative NAAT-RDT result, whereas the opposite was considered a false positive. Sample sizes for each setting were dictated by the number of available participants over the timeframe of the quality initiative. Healthcare workers were trained to administer the NAAT-RDT testing per hospital and the manufacturer's protocols. Sensitivity, specificity, positive predictive value (PPV), and negative predictive value (NPV) were calculated using MedCalc (www.medcalc.org) with 95% confidence intervals (CIs).

## RESULTS

The analytical sensitivity of the Lucira NAAT-RDT was nearly identical to another isothermal NAAT-RDT (i.e., ID NOW) and a LDT, and these three were approximately one 10-fold dilution less sensitive than standard NAATs (i.e., Xpert, Cobas, and Aptima) (Table 1), which was consistent with previous observations for these methods (11). With clinical specimens in the quality initiative, 405 paired tests were evaluated across the three clinical settings. Sensitivity and specificity were consistent between settings (Table 2) with the exception of the pre-operative setting where calculating sensitivity was not possible in the absence of SARS-CoV-2 detections. Overall, sensitivity was 93.0% (95% CI: 83.0%–98.1%), and specificity was 98.3% (95% CI: 96.3%–99.4%) (Table 2).

**TABLE 2** Clinical performance of Lucira Check-It COVID-19 test compared to laboratory-based nucleic acid amplification tests[a,b]

| Setting | Total | TP | TN | FP | FN | Sensitivity (%) (95% CI) | Specificity (%) (95% CI) | PPV (%) (95% CI) | NPV (%) (95% CI) |
|---|---|---|---|---|---|---|---|---|---|
| Pre-operative | 158 | 0 | 156 | 2 | 0 | ND | 98.7 (95.5–99.9) | ND | 100.0 (97.7–100.0) |
| Emergency department | 209 | 49 | 152 | 4 | 4 | 92.5 (81.8–97.9) | 97.4 (93.6–99.3) | 92.5 (82.3–97.0) | 97.4 (93.7–99.0) |
| Community | 38 | 4 | 34 | 0 | 0 | 100.0 (39.8–100.0) | 100.0 (89.7–100.0) | 100.0 (39.8–100.0) | 100.0 (89.7–100.0) |
| All settings | 405 | 53 | 342 | 6 | 4 | 93.0 (83.0–98.1) | 98.3 (96.3–99.4) | 89.8 (79.9–95.1) | 98.8 (97.1–99.6) |

[a]Sensitivity and PPV were unable to be calculated in the pre-operative setting due to the absence of false negative and true positive test results.
[b]CI, confidence interval; FN, false negative; FP, false positive; ND, not determined; NPV, negative predictive value; PPV, positive predictive value; TN, true negative; TP, true positive.

## DISCUSSION

During COVID-19 outbreaks, preserving efficient patient flow and adequate hospital staffing is reliant on streamlined SARS-CoV-2 testing. The time to results for a SARS-CoV-2-positive specimen with a high viral load can be as little as 10 minutes, with negative or positive results from specimens with low viral loads taking up to 30 minutes. Given the transport time required for specimens to clinical laboratories for standard NAAT testing, rapid testing even at 30 minutes is a significant improvement. Quick, reliable, POC testing using the Lucira NAAT-RDT is one way to support patient flow and hospital staffing while facilitating effective infection prevention and control measures and reducing testing demand in clinical laboratories. This NAAT-RDT offered all these advantages, with sensitivity comparable to LDTs used in clinical laboratories (Table 1). While the sensitivity and specificity were slightly lower than standard NAATs used in clinical laboratories (Table 1) (12–15), the performance of this NAAT-RDT was found to be acceptable to expedite SARS-CoV-2 results given it was equivalent to a LDT being used for diagnostic testing (Table 1).

A single other study by Zahavi et al. (4) looked at the real-world performance of the Lucira NAAT-RDT compared to standard NAATs (4) and reported an overall sensitivity of 91.1% (95% CI: 83.2%–96.1%) and specificity of 100.0 (95% CI: 96.4%–100.0%). In their subset of symptomatic individuals, sensitivity was 93.1% (95% CI: 84.5%–97.7%). Both sets of data were nearly identical to those found in our quality initiative, which was primarily focused on asymptomatic testing. We identified six false positive and four false negative NAAT-RDT results. The impact of false positives in the pre-operative setting and ED might delay surgeries or result in improper cohorts of patients. While no false positives were observed in community testing sites, these could lead to delays in staff returning to work. As for the four false negative NAAT-RDT results, two paired specimens had cycle threshold (Ct) values near the level of detection of the laboratory NAATs (34.9 and 36.0), which is consistent with others (4). These low viral loads could represent early infection, remnant RNA from prior infection, or a false positive comparator NAAT (3). Serial testing results over time can be helpful in differentiating between these possibilities but were not available in this quality initiative (6). Standard NAATs are known to be more sensitive than LDTs and NAAT-RDTs using isothermal amplification technologies (3, 12–17); therefore, failure of the Lucira NAAT-RDT to detect SARS-CoV-2 in specimens with low viral loads when compared to these reference tests would not be surprising.

On the other hand, the other two false negative NAAT-RDT results had paired specimens with low Ct values of 18.4 and 18.9 when tested on a standard NAAT, suggesting the specimens had high viral loads seen during the infectious period of SARS-CoV-2. There are multiple possibilities that could explain these discrepant results. To ensure the NAAT-RDT was able to detect specimens with high viral loads, five specimens with Xpert Ct values spanning 16.5–17.5 were tested and were all detected within 10 minutes (data not shown). This excludes the possibility of template inhibition for the NAAT-RDT at high viral loads. Second, contamination could be possible in the clinical laboratory with the NAAT used as the comparator (i.e., Xpert in these two cases) leading to a false positive laboratory NAAT reference value. Given that the individuals with discrepant results between the NAAT-RDT and the standard laboratory NAAT were from different days a month apart, along with the robust quality control practices in clinical laboratories, contamination at high SARS-CoV-2 viral loads during the NAAT comparator testing is extremely unlikely. Another consideration is the timing between NAAT-RDT and standard NAAT testing. In this quality initiative, NAAT-RDT and standard NAAT results were no more than 24 h apart, but the test order was not documented. It is possible that both patients with discrepant results were in an early acute stage of illness detectable in the laboratory by a standard NAAT, but SARS-CoV-2 was not detected with the NAAT-RDT 24 h prior as the viral load was below the level of detection during this period (3). It is also possible that these two false negative NAAT-RDT results represent diagnostic failures, with sequence mismatches occurring between the NAAT-RDT target and circulating SARS-CoV-2; however, this scenario would be unlikely given that the

NAAT-RDT used two different SARS-CoV-2 targets (i.e., N and Orf7b/8 genes), and deleterious mutations would need to occur in both targets to result in a diagnostic failure (18). Mutation may have reduced the sensitivity of NAAT-RDT detection (18). SARS-CoV-2 genomic sequencing should be considered in cases where large discrepancies are observed between two methods; however, no residual specimens were available for further testing in these two cases in our study. Finally, the most likely explanation for the NAAT-RDT false negative results is improper collection or testing inconsistent with manufacturer protocols. Education and training focused on the importance of proper specimen collection, timing of collection, and testing could help avoid possible false negative results and SARS-CoV-2 transmissions. Overall, in the absence of additional testing, the reason for the discrepant results cannot be confirmed in this quality initiative as specimens were not archived for this purpose. Regardless of possible causes, false positive or false negative results are a possibility with any diagnostic test. While we would expect sensitivity and specificity to be nearly 100% for standard NAATs, not all NAATs are created equal (12–15, 19). Here, the performance of the NAAT-RDT was acceptable in the three settings it was implemented. For applications in other settings or testing using new NAAT-RDT formulations (i.e., multiplexed with influenza), independent evaluations should be conducted to assess risks and benefits.

This quality initiative is not without limitations. Specimen numbers were low in the community setting, and no positives were obtained in the pre-operative setting. Additional comparative studies could be useful to strengthen the data for these settings. Next, the status of individuals as symptomatic or asymptomatic in the emergency department was also not available. Another main limitation of this quality initiative is the absence of data available to assess the cost–benefits and impact of NAAT-RDT implementation in the testing settings. Cost analyses were not possible given that NAAT-RDT kits and NAAT reagents and consumables for clinical testing were provided by provincial and federal initiatives to support SARS-CoV-2 testing. As for the assessment of impact, data from other studies using another NAAT-RDT show that rapid results facilitate clinical decision-making, improve the use of recommended treatments for COVID-19, and streamline patient flow (7). In the pre-operative setting, POC testing allowed the rescheduling of elective surgeries of asymptomatic patients who test positive for SARS-CoV-2, avoiding the risk of post-surgical morbidity and mortality or potential nosocomial transmissions (8). POC testing was also shown to increase a sense of workplace safety and improve morale (9) and was easily incorporated into pre-operative and occupational health decision-making (9–11). Future studies should consider evaluating the impact of NAAT-RDTs in each setting, comparing individuals who present with and without symptoms, as well as considerations for costs compared to laboratory testing.

Overall, this quality initiative demonstrated that the Lucira NAAT-RDT offers a feasible option for rapid and portable SARS-CoV-2 testing, with comparable performance to laboratory NAATs. While this NAAT-RDT should be validated for each local application, it can be employed with little training for specimen collection, processing, and interpretation, and likely at a cost justifiable from the resources saved from avoiding sample transport and laboratory testing.

## ACKNOWLEDGMENTS

All testing was performed as a standard of care at Nova Scotia Health.

The authors would like to acknowledge the hard work of the bedside clinical staff who collected the Lucira specimens and recorded the test results used in this quality initiative. The authors would also like to recognize the tireless efforts of the laboratory staff that support SARS-CoV-2 testing initiatives throughout the pandemic.

This research did not receive any specific grant from funding agencies in the public, commercial, or not-for-profit sectors.

E.S. performed the conceptualization, methodology, validation, formal analysis, investigation, data curation, writing (original draft), writing (review and editing),

visualization, supervision, and project administration. G.R.M. performed the conceptualization, methodology, validation, formal analysis, investigation, and writing (review and editing). T.F.H. performed the conceptualization, methodology, validation, investigation, writing (original draft), writing (review and editing), supervision, and project administration. S.A.M. performed the conceptualization, writing (review and editing), supervision, and project administration. I.D. performed the conceptualization, writing (review and editing), supervision, and project administration. N.W. performed the conceptualization and writing (review and editing). A.K. performed the conceptualization and writing (review and editing). J.J.L. performed the conceptualization, methodology, validation, formal analysis, investigation, data curation, writing (original draft), writing (review and editing), visualization, supervision, and project administration. G.P. performed the conceptualization, methodology, validation, formal analysis, investigation, data curation, writing (original draft, writing (review and editing), visualization, supervision, and project administration.

The authors declare that they have no known competing financial interests or personal relationships that could have appeared to influence the work reported in this paper.

## AUTHOR AFFILIATIONS

[1]Department of Medicine, Faculty of Medicine, Dalhousie University, Halifax, Nova Scotia, Canada
[2]Department of Pathology, Faculty of Medicine, Dalhousie University, Halifax, Nova Scotia, Canada
[3]Division of Microbiology, Department of Pathology and Laboratory Medicine, Nova Scotia Health Authority, Halifax, Nova Scotia, Canada
[4]COVID-19 Implementation and Planning, Nova Scotia Health, Halifax, Nova Scotia, Canada
[5]Occupational Health Safety & Wellness, People Services, Nova Scotia Health, Halifax, Nova Scotia, Canada

## AUTHOR ORCIDs

Elizabeth Simms http://orcid.org/0009-0003-7864-8595
Jason J. LeBlanc http://orcid.org/0000-0003-0593-0357
Glenn Patriquin http://orcid.org/0000-0001-8674-4358

## ETHICS APPROVAL

This assessment was deemed a Quality Initiative Project by the Nova Scotia Health Research Ethics Board (file number 1027683) and was therefore exempt from full review. Data were collected and analyzed with intent to ensure quality of local practice.

## ADDITIONAL FILES

The following material is available online.

Open Peer Review

**PEER REVIEW HISTORY (review-history.pdf).** An accounting of the reviewer comments and feedback.

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
