## [Reviewer comments · Microbiology Spectrum]

Microbiology Spectrum

Real-world evaluation of the Lucira Check-It COVID-19 loop-mediated amplification (LAMP) test

Elizabeth Simms, Gregory McCracken, Todd Hatchette, Shelly McNeil, Ian Davis, Noella Whelan, Angela Keenan, Jason LeBlanc, and Glenn Patriquin

Corresponding Author(s): Elizabeth Simms, Dalhousie University

Review Timeline:

Submission Date:	July 20, 2023
Editorial Decision:	August 8, 2023
Revision Received:	August 30, 2023
Editorial Decision:	September 11, 2023
Revision Received:	September 27, 2023
Accepted:	September 29, 2023

Editor: Tulip Jhaveri

Reviewer(s): Disclosure of reviewer identity is with reference to reviewer comments included in decision letter(s). The following individuals involved in review of your submission have agreed to reveal their identity: Xianding Deng (Reviewer #1)

Transaction Report:

DOI: <https://doi.org/10.1128/spectrum.02772-23>

August 8, 2023

Dr. Elizabeth Simms
Dalhousie University
Department of Medicine, Department of Pathology
5820 University Avenue
Dickson Building, 5th floor
Halifax, Nova Scotia B3H2Y9
Canada

Re: Spectrum02772-23 (Real-world evaluation of the Lucira Check-It COVID-19 loop-mediated amplification (LAMP) test)

Dear Dr. Elizabeth Simms:

Thank you for submitting your manuscript to Microbiology Spectrum. Major revisions are needed before this manuscript can be considered for publication. When submitting the revised version of your paper, please provide (1) point-by-point responses to the issues raised by the reviewers as file type "Response to Reviewers," not in your cover letter, and (2) a PDF file that indicates the changes from the original submission (by highlighting or underlining the changes) as file type "Marked Up Manuscript - For Review Only". Please use this link to submit your revised manuscript - we strongly recommend that you submit your paper within the next 60 days or reach out to me. Detailed instructions on submitting your revised paper are below.

Link Not Available

Sincerely,

Tulip Jhaveri

Journals Department
Reviewer comments:

Reviewer #1 (Comments for the Author):

Authors evaluated Lucira COVID-19 LAMP test performance in hospital and clinic settings compared to standard NAAT. Though the results look promising, the paper miss lots of data (only 1 table included).

In Background, authors should expand NAAT-based RDTs; as far as I know, there are diverse rapid methods for SARS-CoV-2 testing, such as ID NOW, LAMP-CRISPR, SDA, RPA. Authors should also give some background of Lucira, e.g. mechanism of the assay, is the readout colorimetric or fluorescence?

Can authors elaborate how many samples were tested by each of Cobas, Xpert and Altima platforms, were all 405 samples tested on these three qPCR assays to derive concordance? We know there are variations of performance between platforms. Maybe a table should be included to describe how concordance of TP and TN is obtained.

Authors should include a table or figure showing how the serial dilutions were made, where is the result of 10-fold dilutions compared to ID NOW and other assays, what is the limited of detection of Lucira?

LAMP might have cross-contamination issues, how did authors address that; are those FP due to cross-contamination?

What are the disadvantages of LAMP-based Lucira assay, will the time-to-result get longer for low-titer virus samples?

What about the cost of Lucira assay compared to other rapid NAAT assays in terms of cost per sample, is it significantly cheaper?

Reviewer #2 (Comments for the Author):

See attachment..

Staff Comments:

Preparing Revision Guidelines

Please return the manuscript within 60 days; if you cannot complete the modification within this time period, please contact me. If you do not wish to modify the manuscript and prefer to submit it to another journal, please notify me of your decision immediately so that the manuscript may be formally withdrawn from consideration by Microbiology Spectrum.

Comment 1:

- The authors should provide an explanation in Table 1 for the absence of determined sensitivity in the perioperative diagnostic test and possibly include the PPV as well.
- I am unsure about the meaning of "perioperative" as used by the authors. Using the term "pre-operative screening" might be more appropriate.
- The sensitivity of the diagnostic tests in the emergency department was not sufficiently high. This aspect requires constructive criticism and a thorough examination of its limitations.
- The authors asserted that the majority of patients from both the pre-operative screening and emergency conditions were asymptomatic (lacking respiratory symptoms). If the authors concur, they should provide the rationale behind this observation and the pretest likelihood ratio and engage in further discussion on this matter.
- Table 1 should include the PPV and NPV values.

Comment 2:

The authors should explain the sample size calculation for all settings, but not for only one of perioperative, ED, and community cases. Additionally, to improve clarity, all settings groups in Table 1 should be displayed in the first row of the table.

Comment 3:

Both the abstract and manuscript should be written concisely using well-refined English to enhance overall readability and quality.

Comment 4:

The authors should offer clear definitions of NAAT (Nucleic Acid Amplification Test) and NAAT-RDT (Nucleic Acid Amplification Test - Rapid Diagnostic Test) and consistently use these defined terms throughout the entire manuscript.

Comment 5:

The authors should provide a comprehensive definition of asymptomatic close contacts of COVID-19 cases and elucidate the reasons why evaluating these patients is essential for the study's objectives.

Comment 6:

The authors should provide information on the Cycle threshold (Ct) value of NAAT in true positive (TP) and true negative (TN) results of NAAT-RDT for better understanding and analysis.

Comment 7:

Regarding the claim of real-world testing, it is essential for the authors to specify the total number of both symptomatic and asymptomatic patients who underwent NAAT-RDT testing to provide a clearer context for the study's applicability.

Comment 8:

The implementation of this study title seems to focus on asymptomatic individuals rather than asymptomatic individuals suspected to have COVID-19. However, it would be beneficial if the authors could present more data regarding the sensitivity and specificity in symptomatic COVID-19 cases. These data should be cited in the introduction and appropriately referenced.

Comment 9:

In Line 130, the authors mentioned that the other two false negative results from the Lucira NAAT-RDT had paired specimens with low Ct values of 18.4 and 18.9, indicating high viral loads during the infectious period of SARS-CoV-2. The authors should elaborate on why negative NAAT-RDT results can occur during a period of high viral load to provide a more comprehensive understanding of the test's performance under such circumstances.

Comment 10:

In Line 134, the authors suggest that false negative Lucira results may be due to diagnostic failures resulting from sequence mismatches between the Lucira target and circulating SARS-CoV-2. However, to provide a more comprehensive understanding, the introduction should include additional details about the basis and principles of the Lucira test, particularly regarding the gene detection part before discussing the possibility of needing to resort to sequencing.

Comment 11:

In Line 137, the authors acknowledge that false positive (FP) and false negative (FN) results can occur with any diagnostic test. It is important to note that while these occurrences are possible for any test, for nucleic acid amplification tests (NAATs) in general, higher sensitivity and specificity are usually expected, often nearing 100%.

Comment 12:

The authors should expand on the limitations of the study to provide a more comprehensive assessment of potential constraints or factors that might affect the interpretation of the results. Addressing these limitations will contribute to a more well-rounded discussion of the study's findings.

Spectrum Reviewer Comments

Reviewer 1

Authors evaluated Lucira COVID-19 LAMP test performance in hospital and clinic settings compared to standard NAAT. Though the results look promising, the paper miss lots of data (only 1 table included). Supplementary Table S1 that contained relevant data was not cited in the body of the text. We apologize for this oversight. Table S1 has been moved from the supplementary material to the body of the manuscript to make it more visible to readers and we have re-labelled it Table 1 (and re-labelled the previous Table 1 as Table 2 accordingly). Unfortunately, we are limited beyond this by two figure/table limitations of the “Observations” article type.

In Background, authors should expand NAAT-based RDTs; as far as I know, there are diverse rapid methods for SARS-CoV-2 testing, such as ID NOW, LAMP-CRISPR, SDA, RPA. Authors should also give some background of Lucira, e.g. mechanism of the assay, is the readout colorimetric or fluorescence? We have modified the background to recognize the other isothermal technologies and indicated the rationale of why Lucira was chosen. We have also added more to the description of the technology used with Lucira, a RT-LAMP targeting N and Orf7b/8 genes with a colorimetric output detected by change in pH during target amplification. See lines 73-76.

Can authors elaborate how many samples were tested by each of Cobas, Xpert and Altima platforms, were all 405 samples tested on these three qPCR assays to derive concordance? We know there are variations of performance between platforms. Maybe a table should be included to describe how concordance of TP and TN is obtained.

NAAT results reported to the ordering physician did not distinguish which diagnostic test was used and therefore, the number for each NAAT was not possible to collect for this quality initiative. The Lucira results were compared to one of three possible commercial NAATs used in clinical laboratories. This has been clarified in the text. How the NAAT results were used to define TP and TN Lucira was defined in the methods. We have added modifications to clarify the methods. See lines 103-112.

Authors should include a table or figure showing how the serial dilutions were made, where is the result of 10-fold dilutions compared to ID NOW and other assays, what is the limited of detection of Lucira?

Table 1 (formerly Table S1) has been moved into the body of the manuscript to show this data and the methods describe the serial dilutions.

LAMP might have cross-contamination issues, how did authors address that; are those FP due to cross-contamination?

Lucira testing was administered in clinical settings in the closed portable instrument, and therefore cross-contamination from other specimens or from prior amplification would be unlikely. This sentence has been added to the discussion on lines 155-159.

What are the disadvantages of LAMP-based Lucira assay, will the time-to-result get longer for low-titer virus samples?

Time to results for a specimen with a high viral load can be as little as 10 min while negative or positive results from specimens with low viral loads take up to 30 min. Given the transport time to labs to perform NAAT testing, quick testing even at (~30 min) is a significant improvement. Other benefits of Lucira and limitations have been added to the discussion.

What about the cost of Lucira assay compared to other rapid NAAT assays in terms of cost per sample, is it significantly cheaper?

Cost analyses were not possible given that Lucira kits, NAAT reagents, and consumables for clinical testing were provided by provincial and federal initiatives to support SARS-CoV-2 testing. However, this was added as a study limitation. See lines 189-191.

Reviewer 2

Comment 1:

• The authors should provide an explanation in Table 1 for the absence of determined sensitivity in the perioperative diagnostic test and possibly include the PPV as well.

Sensitivity and PPV was unable to be calculated due to an absence of false negative tests and true positive tests in the pre-operative setting. This was added to the Table 2 footnote (formerly Table 1).

• I am unsure about the meaning of "perioperative" as used by the authors. Using the term "pre-operative screening" might be more appropriate.

Perioperative was changed to pre-operative throughout the manuscript.

• The sensitivity of the diagnostic tests in the emergency department was not sufficiently high. This aspect requires constructive criticism and a thorough examination of its limitations.

We have added a sentence in the discussion to justify the rationale that the sensitivity of Lucira was deemed acceptable. See lines 132-134: "While the sensitivity and specificity were slightly lower than commercial NAATs used in clinical laboratories (Table 1) [12-15], the performance of Lucira was found to be acceptable to expedite SARS-CoV-2 results given it was equivalent to a LDT being used for diagnostic testing (Table 1)."

• The authors asserted that the majority of patients from both the pre-operative screening and emergency conditions were asymptomatic (lacking respiratory symptoms). If the authors concur, they should provide the rationale behind this observation and the pretest likelihood ratio and engage in further discussion on this matter.

Data on patients presenting to emergency departments (symptomatic or asymptomatic) was not available, and this has been added as a study limitation. Investigating differences between symptomatic and asymptomatic individuals was added to the study future direction. See lines 186-187.

Table 1 should include the PPV and NPV values.

PPV and NPV were added to Table 2 (formerly Table 1), when possible.

Comment 2:

The authors should explain the sample size calculation for all settings, but not for only one of perioperative, ED, and community cases. Additionally, to improve clarity, all settings groups in Table 1 should be displayed in the first row of the table.

Sample sizes for each setting were dictated by the number of available participants over the timeframe of the quality initiative.

Comment 3:

Both the abstract and manuscript should be written concisely using well-refined English to enhance overall readability and quality.

English and grammar has been reviewed throughout the manuscript.

Comment 4:

The authors should offer clear definitions of NAAT (Nucleic Acid Amplification Test) and NAAT-RDT (Nucleic Acid Amplification Test - Rapid Diagnostic Test) and consistently use these defined terms throughout the entire manuscript.

These terms were defined in the background section (lines 67-72) and have been used consistently thereafter.

Comment 5:

The authors should provide a comprehensive definition of asymptomatic close contacts of COVID-19 cases and elucidate the reasons why evaluating these patients is essential for the study's objectives.

Sentences were added to explain that healthcare workers with close contacts to positive SARS-CoV-2 cases were instructed to remain home for seven days following exposure. Asymptomatic testing was implemented to help these individuals return to work if test results were negative, reducing concerns of subsequent transmission.

Comment 6:

The authors should provide information on the Cycle threshold (Ct) value of NAAT in true positive (TP) and true negative (TN) results of NAAT-RDT for better understanding and analysis.

These values were provided in the discussion with possible explanations.

Comment 7:

Regarding the claim of real-world testing, it is essential for the authors to specify the total number of both symptomatic and asymptomatic patients who underwent NAAT-RDT testing to provide a clearer context for the study's applicability.

All patients in the pre-operative and close contact/HCW groups were asymptomatic as described in the methods. Based on the nature of this quality improvement project, it is not possible to determine symptom status of the ED patients which was described but has now been re-emphasized as a limitation in the discussion.

Comment 8:

The implementation of this study title seems to focus on asymptomatic individuals rather than asymptomatic individuals suspected to have COVID-19. However, it would be beneficial if the authors could present more data regarding the sensitivity and specificity in symptomatic COVID-19 cases. These data should be cited in the introduction and appropriately referenced.

Comparisons between the performance of Lucira in symptomatic and asymptomatic individuals was found in the discussion, but we have better emphasized this topic in comparison to the only other study who reported Lucira test performance data (lines 135-139): "A single other study by Zahavi et al. [4] looked at the real-world performance of Lucira compared to NAATs [4], and the reported an overall sensitivity of 91.1% (95% CI: 83.2 to 96.1%) and specificity of 100.0 (95% CI: 96.4% to 100.0%). In their subset of symptomatic individuals, sensitivity was 93.1% (95% CI: 84.5% to 97.7%). Both sets of data were nearly identical to those found in our quality initiative which was primarily focused on asymptomatic testing."

Comment 9:

In Line 130, the authors mentioned that the other two false negative results from the Lucira NAAT-RDT had paired specimens with low Ct values of 18.4 and 18.9, indicating high viral loads during the infectious period of SARS-CoV-2. The authors should elaborate on why negative NAAT-RDT results can

occur during a period of high viral load to provide a more comprehensive understanding of the test's performance under such circumstances.

We have added to the discussion many possible factors that could explain the two discrepant results.

Comment 10:

In Line 134, the authors suggest that false negative Lucira results may be due to diagnostic failures resulting from sequence mismatches between the Lucira target and circulating SARS-CoV-2. However, to provide a more comprehensive understanding, the introduction should include additional details about the basis and principles of the Lucira test, particularly regarding the gene detection part before discussing the possibility of needing to resort to sequencing.

The background has been modified to add more about the principle of Lucira, and the discussion has revised the content for sequencing and added a reference to support the situations where sequencing should be considered.

Comment 11:

In Line 137, the authors acknowledge that false positive (FP) and false negative (FN) results can occur with any diagnostic test. It is important to note that while these occurrences are possible for any test, for nucleic acid amplification tests (NAATs) in general, higher sensitivity and specificity are usually expected, often nearing 100%.

This has been elaborated on in the discussion.

Comment 12:

The authors should expand on the limitations of the study to provide a more comprehensive assessment of potential constraints or factors that might affect the interpretation of the results. Addressing these limitations will contribute to a more well-rounded discussion of the study's findings. Additional limitations and future directions have been added to the discussion.

September 11, 2023

Dr. Elizabeth Simms
Dalhousie University
Department of Medicine, Department of Pathology
5820 University Avenue
Dickson Building, 5th floor
Halifax, Nova Scotia B3H2Y9
Canada

Re: Spectrum02772-23R1 (Real-world evaluation of the Lucira Check-It COVID-19 loop-mediated amplification (LAMP) test)

Dear Dr. Elizabeth Simms:

Link Not Available

Sincerely,

Tulip Jhaveri

Journals Department
Reviewer comments:

Reviewer #1 (Comments for the Author):

my comments and questions have been addressed.

Reviewer #2 (Comments for the Author):

Comment 1:

The author should be asked to provide reference values for SARS-CoV-2 concentrations in Table 1 that are correlated with the Lucira Ct values, especially when using a standard virus for measurement. This would enhance the clarity of the data.

Comment 2:

It's recommended to include the sentence "majority of asymptomatic settings in WHO screening in pre-operative and community" somewhere in the text for context.

Comment 3:

In Table 2, there seems to be confusion when calculating TP, TN, FP, FN in each setting, leading to incorrect numbers. The authors should refer to the 2x2 table to ensure accurate numbers and adhere to the STARD guideline for reporting. Careful review of the results is necessary.

Comment 4

Line 107: True positive and true negative results were defined by concordant results between Lucira and one of the three commercial NAATs.

To ensure clarity and reliability, it's advisable to choose a single reference standard PCR test and provide detailed information about this test, including the cutoff value used. This will establish a clear benchmark for comparison in the study and enhance the validity of the results.

Comment 5

Please provide a flowchart of participant enrollment and characteristics according to the STARD guideline to enhance clarity.

Comment 6

In the discussion section, it would be beneficial to present the main results or conclusions before delving into details.

Comment 7

Reduce the use of the term "Lucira" and consider using terms like "NAAT-RDT" or "POC testing" instead for variety and clarity.

Comment 8

Line 132: While the sensitivity and specificity were slightly lower than commercial NAATs used in clinical laboratories (Table 1) Certainly, emphasizing the term "standard NAATs" over "commercial NAATs" in the discussion and throughout the paper would provide greater clarity and alignment with established testing practices.

Comment 9

Line 146: commercial NAATs are known to be more sensitive than LDTs and NAAT-RDTs using isothermal amplification technologies [3, 12-17], therefore failure of Lucira to detect SARS-CoV-2 in specimens with low viral loads was not surprising. I suggest the authored remove and replace with interpretation comparing with the standard reference test.

Comment 10

Line 150: On the other hand, the other two false negative Lucira results had paired specimens with low Ct values of 18.4 and 18.9 when tested on a commercial NAAT, suggesting the specimens had high viral loads seen during the infectious period of SARS-CoV-2. There are multiple possibilities that could explain these discrepant results. To ensure Lucira was able to detect specimens with high viral loads, 5 specimens with Xpert Ct values spanning 16.5 to 17.5 were tested and were all detected within 10 minutes (data not shown).

I should in this point the author should carefully discuss and provide the result in supplementary.

Comment 11

Line 155: Please remove "Second, Lucira testing was administered at the point-of-care in with the closed portable single-use instrument, and therefore cross-contamination between specimens or from prior testing in the same location would be unlikely; however, contamination could be possible in a clinical laboratory with the NAAT used as the comparator (i.e., Xpert in these two cases)."

Your suggestion to carefully interpret the findings and review potential sources of bias, including pre-analytical, analytical, and post-analytical factors, is valid. This thorough examination of errors and biases can help to better understand the reliability of the study and the selection of the standard test. It's important to consider and address these factors in the interpretation of the results for a more comprehensive and accurate assessment.

Comment 12

Line 164: It is possible that both patients with discrepant result were in an early acute stage of illness detectable in the laboratory by a commercial NAAT, but SARS-CoV-2 was not detected with Lucira 24h prior as the viral load was below the level of detection.

Your point regarding the need for a detailed comparison of methods and processing between Lucira and standard NAATs is valid. If there are differences in specimen collection, handling, or processing, these could indeed contribute to false negatives or other discrepancies in results. Addressing these differences and their potential impact on the study's findings would enhance the overall reliability and interpretation of the results. It's important for the authors to provide a comprehensive understanding of the testing procedures to assess potential sources of error and bias.

Comment 13

Education and training focused on the importance of proper specimen collection, timing of collection, and testing could help avoid possible false negative results and SARS-CoV-2 transmissions
The authors should declare this training and method in the early of manuscript.

Comment 14

Line 187: Another main limitation of this quality initiative is the absence of data available to assess the cost-benefits and impact of Lucira implementation in the testing settings. Cost analyses were not possible given Lucira kits and NAAT reagents and consumables for clinical testing were provided by provincial and federal initiatives to support SARS-CoV-2 testing. I think this issue did not relate to this topic.

Comment 15:

The language in limitation part should be revised.

Staff Comments:

Preparing Revision Guidelines

Please return the manuscript within 60 days; if you cannot complete the modification within this time period, please contact me. If you do not wish to modify the manuscript and prefer to submit it to another journal, please notify me of your decision immediately so that the manuscript may be formally withdrawn from consideration by Microbiology Spectrum.

Spectrum Reviewer Comments

Reviewer 1

Authors evaluated Lucira COVID-19 LAMP test performance in hospital and clinic settings compared to standard NAAT. Though the results look promising, the paper miss lots of data (only 1 table included). Supplementary Table S1 that contained relevant data was not cited in the body of the text. We apologize for this oversight. Table S1 has been moved from the supplementary material to the body of the manuscript to make it more visible to readers and we have re-labelled it Table 1 (and re-labelled the previous Table 1 as Table 2 accordingly). Unfortunately, we are limited beyond this by two figure/table limitations of the “Observations” article type.

In Background, authors should expand NAAT-based RDTs; as far as I know, there are diverse rapid methods for SARS-CoV-2 testing, such as ID NOW, LAMP-CRISPR, SDA, RPA. Authors should also give some background of Lucira, e.g. mechanism of the assay, is the readout colorimetric or fluorescence? We have modified the background to recognize the other isothermal technologies and indicated the rationale of why Lucira was chosen. We have also added more to the description of the technology used with Lucira, a RT-LAMP targeting N and Orf7b/8 genes with a colorimetric output detected by change in pH during target amplification. See lines 73-76.

Can authors elaborate how many samples were tested by each of Cobas, Xpert and Altima platforms, were all 405 samples tested on these three qPCR assays to derive concordance? We know there are variations of performance between platforms. Maybe a table should be included to describe how concordance of TP and TN is obtained.

NAAT results reported to the ordering physician did not distinguish which diagnostic test was used and therefore, the number for each NAAT was not possible to collect for this quality initiative. The Lucira results were compared to one of three possible commercial NAATs used in clinical laboratories. This has been clarified in the text. How the NAAT results were used to define TP and TN Lucira was defined in the methods. We have added modifications to clarify the methods. See lines 103-112.

Authors should include a table or figure showing how the serial dilutions were made, where is the result of 10-fold dilutions compared to ID NOW and other assays, what is the limited of detection of Lucira?

Table 1 (formerly Table S1) has been moved into the body of the manuscript to show this data and the methods describe the serial dilutions.

LAMP might have cross-contamination issues, how did authors address that; are those FP due to cross-contamination?

Lucira testing was administered in clinical settings in the closed portable instrument, and therefore cross-contamination from other specimens or from prior amplification would be unlikely. This sentence has been added to the discussion on lines 155-159.

What are the disadvantages of LAMP-based Lucira assay, will the time-to-result get longer for low-titer virus samples?

Time to results for a specimen with a high viral load can be as little as 10 min while negative or positive results from specimens with low viral loads take up to 30 min. Given the transport time to labs to perform NAAT testing, quick testing even at (~30 min) is a significant improvement. Other benefits of Lucira and limitations have been added to the discussion.

What about the cost of Lucira assay compared to other rapid NAAT assays in terms of cost per sample, is it significantly cheaper?

Cost analyses were not possible given that Lucira kits, NAAT reagents, and consumables for clinical testing were provided by provincial and federal initiatives to support SARS-CoV-2 testing. However, this was added as a study limitation. See lines 189-191.

Reviewer 2

Comment 1:

• The authors should provide an explanation in Table 1 for the absence of determined sensitivity in the perioperative diagnostic test and possibly include the PPV as well.

Sensitivity and PPV was unable to be calculated due to an absence of false negative tests and true positive tests in the pre-operative setting. This was added to the Table 2 footnote (formerly Table 1).

• I am unsure about the meaning of "perioperative" as used by the authors. Using the term "pre-operative screening" might be more appropriate.

Perioperative was changed to pre-operative throughout the manuscript.

• The sensitivity of the diagnostic tests in the emergency department was not sufficiently high. This aspect requires constructive criticism and a thorough examination of its limitations.

We have added a sentence in the discussion to justify the rationale that the sensitivity of Lucira was deemed acceptable. See lines 132-134: "While the sensitivity and specificity were slightly lower than commercial NAATs used in clinical laboratories (Table 1) [12-15], the performance of Lucira was found to be acceptable to expedite SARS-CoV-2 results given it was equivalent to a LDT being used for diagnostic testing (Table 1)."

• The authors asserted that the majority of patients from both the pre-operative screening and emergency conditions were asymptomatic (lacking respiratory symptoms). If the authors concur, they should provide the rationale behind this observation and the pretest likelihood ratio and engage in further discussion on this matter.

Data on patients presenting to emergency departments (symptomatic or asymptomatic) was not available, and this has been added as a study limitation. Investigating differences between symptomatic and asymptomatic individuals was added to the study future direction. See lines 186-187.

Table 1 should include the PPV and NPV values.

PPV and NPV were added to Table 2 (formerly Table 1), when possible.

Comment 2:

The authors should explain the sample size calculation for all settings, but not for only one of perioperative, ED, and community cases. Additionally, to improve clarity, all settings groups in Table 1 should be displayed in the first row of the table.

Sample sizes for each setting were dictated by the number of available participants over the timeframe of the quality initiative.

Comment 3:

Both the abstract and manuscript should be written concisely using well-refined English to enhance overall readability and quality.

English and grammar has been reviewed throughout the manuscript.

Comment 4:

The authors should offer clear definitions of NAAT (Nucleic Acid Amplification Test) and NAAT-RDT (Nucleic Acid Amplification Test - Rapid Diagnostic Test) and consistently use these defined terms throughout the entire manuscript.

These terms were defined in the background section (lines 67-72) and have been used consistently thereafter.

Comment 5:

The authors should provide a comprehensive definition of asymptomatic close contacts of COVID-19 cases and elucidate the reasons why evaluating these patients is essential for the study's objectives.

Sentences were added to explain that healthcare workers with close contacts to positive SARS-CoV-2 cases were instructed to remain home for seven days following exposure. Asymptomatic testing was implemented to help these individuals return to work if test results were negative, reducing concerns of subsequent transmission.

Comment 6:

The authors should provide information on the Cycle threshold (Ct) value of NAAT in true positive (TP) and true negative (TN) results of NAAT-RDT for better understanding and analysis.

These values were provided in the discussion with possible explanations.

Comment 7:

Regarding the claim of real-world testing, it is essential for the authors to specify the total number of both symptomatic and asymptomatic patients who underwent NAAT-RDT testing to provide a clearer context for the study's applicability.

All patients in the pre-operative and close contact/HCW groups were asymptomatic as described in the methods. Based on the nature of this quality improvement project, it is not possible to determine symptom status of the ED patients which was described but has now been re-emphasized as a limitation in the discussion.

Comment 8:

The implementation of this study title seems to focus on asymptomatic individuals rather than asymptomatic individuals suspected to have COVID-19. However, it would be beneficial if the authors could present more data regarding the sensitivity and specificity in symptomatic COVID-19 cases. These data should be cited in the introduction and appropriately referenced.

Comparisons between the performance of Lucira in symptomatic and asymptomatic individuals was found in the discussion, but we have better emphasized this topic in comparison to the only other study who reported Lucira test performance data (lines 135-139): "A single other study by Zahavi et al. [4] looked at the real-world performance of Lucira compared to NAATs [4], and the reported an overall sensitivity of 91.1% (95% CI: 83.2 to 96.1%) and specificity of 100.0 (95% CI: 96.4% to 100.0%). In their subset of symptomatic individuals, sensitivity was 93.1% (95% CI: 84.5% to 97.7%). Both sets of data were nearly identical to those found in our quality initiative which was primarily focused on asymptomatic testing."

Comment 9:

In Line 130, the authors mentioned that the other two false negative results from the Lucira NAAT-RDT had paired specimens with low Ct values of 18.4 and 18.9, indicating high viral loads during the infectious period of SARS-CoV-2. The authors should elaborate on why negative NAAT-RDT results can

occur during a period of high viral load to provide a more comprehensive understanding of the test's performance under such circumstances.

We have added to the discussion many possible factors that could explain the two discrepant results.

Comment 10:

In Line 134, the authors suggest that false negative Lucira results may be due to diagnostic failures resulting from sequence mismatches between the Lucira target and circulating SARS-CoV-2. However, to provide a more comprehensive understanding, the introduction should include additional details about the basis and principles of the Lucira test, particularly regarding the gene detection part before discussing the possibility of needing to resort to sequencing.

The background has been modified to add more about the principle of Lucira, and the discussion has revised the content for sequencing and added a reference to support the situations where sequencing should be considered.

Comment 11:

In Line 137, the authors acknowledge that false positive (FP) and false negative (FN) results can occur with any diagnostic test. It is important to note that while these occurrences are possible for any test, for nucleic acid amplification tests (NAATs) in general, higher sensitivity and specificity are usually expected, often nearing 100%.

This has been elaborated on in the discussion.

Comment 12:

The authors should expand on the limitations of the study to provide a more comprehensive assessment of potential constraints or factors that might affect the interpretation of the results. Addressing these limitations will contribute to a more well-rounded discussion of the study's findings. Additional limitations and future directions have been added to the discussion.

Response to Reviewer Comments

Comment 1:

The author should be asked to provide reference values for SARS-CoV-2 concentrations in Table 1 that are correlated with the Lucira Ct values, especially when using a standard virus for measurement. This would enhance the clarity of the data.

Please see the first column of Table 1 for reference values of SARS-CoV-2 concentrations in copies/mL.

Comment 2:

It's recommended to include the sentence "majority of asymptomatic settings in WHO screening in pre-operative and community" somewhere in the text for context.

The authors are not familiar with WHO screening recommendations/criteria in his context. We would request the reviewer clarify this or provide a resource for us to consult.

Comment 3:

In Table 2, there seems to be confusion when calculating TP, TN, FP, FN in each setting, leading to incorrect numbers. The authors should refer to the 2x2 table to ensure accurate numbers and adhere to the STARD guideline for reporting. Careful review of the results is necessary.

The results displayed in Table 2 have been reviewed. No errors were found.

Comment 4

Line 107: True positive and true negative results were defined by concordant results between Lucira and one of the three commercial NAATs.

To ensure clarity and reliability, it's advisable to choose a single reference standard PCR test and provide detailed information about this test, including the cutoff value used. This will establish a clear benchmark for comparison in the study and enhance the validity of the results.

In order to maximize laboratory efficiency while processing high sample volumes during the SARS-CoV-2 pandemic, several standard NAATs were used in our lab at the time of data collection. As outlined in Table 1, results were comparable amongst NAATs used.

Comment 5

Please provide a flowchart of participant enrollment and characteristics according to the STARD guideline to enhance clarity.

This quality initiative did not collect any data on participant characteristics.

Comment 6

In the discussion section, it would be beneficial to present the main results or conclusions before delving into details.

Please see paragraph one of the discussion section for main results and conclusions (lines 130-137).

Comment 7

Reduce the use of the term "Lucira" and consider using terms like "NAAT-RDT" or "POC testing" instead for variety and clarity.

This edit has been made in the manuscript.

Comment 8

Line 132: While the sensitivity and specificity were slightly lower than commercial NAATs used in clinical laboratories (Table 1)

Certainly, emphasizing the term "standard NAATs" over "commercial NAATs" in the discussion and throughout the paper would provide greater clarity and alignment with established testing practices.

This edit has been made in the manuscript.

Comment 9

Line 146: commercial NAATs are known to be more sensitive than LDTs and NAAT-RDTs using isothermal amplification technologies [3, 12-17], therefore failure of Lucira to detect SARS-CoV-2 in specimens with low viral loads was not surprising.

I suggest the authored remove and replace with interpretation comparing with the standard reference test.

This line has been edited.

Comment 10

Line 150: On the other hand, the other two false negative Lucira results had paired specimens with low Ct values of 18.4 and 18.9 when tested on a commercial NAAT, suggesting the specimens had high viral loads seen during the infectious period of SARS-CoV-2. There are multiple possibilities that could explain these discrepant results. To ensure Lucira was able to detect specimens with high viral loads, 5 specimens with Xpert Ct values spanning 16.5 to 17.5 were tested and were all detected within 10 minutes (data not shown).

I should in this point the author should carefully discuss and provide the result in supplementary.

The purpose of performing this additional testing was to exclude template inhibition as a possible reason for false negative tests. This is described in the section where false negatives are discussed.

Comment 11

Line 155: Please remove "Second, Lucira testing was administered at the point-of-care in with the closed portable single-use instrument, and therefore cross-contamination between specimens or from prior testing in the same location would be unlikely; however, contamination could be possible in a clinical laboratory with the NAAT used as the comparator (i.e., Xpert in these two cases)."

Your suggestion to carefully interpret the findings and review potential sources of bias, including pre-analytical, analytical, and post-analytical factors, is valid. This thorough examination of errors and biases can help to better understand the reliability of the study and the selection of the standard test. It's important to consider and address these factors in the interpretation of the results for a more comprehensive and accurate assessment.

We have reworded this section to recognize the possibility of false positive or false negative results occurring from either the NAAT-RDT or the standard NAATs in the laboratory.

Comment 12

Line 164: It is possible that both patients with discrepant result were in an early acute stage of illness detectable in the laboratory by a commercial NAAT, but SARS-CoV-2 was not detected with Lucira 24h prior as the viral load was below the level of detection.

Your point regarding the need for a detailed comparison of methods and processing between Lucira and standard NAATs is valid. If there are differences in specimen collection, handling, or processing, these could indeed contribute to false negatives or other discrepancies in results. Addressing these differences and their potential impact on the study's findings would enhance the overall reliability and interpretation of the results. It's important for the authors to provide a comprehensive understanding of the testing procedures to assess potential sources of error and bias.

The potential of improper collection, handling, or processing was mentioned in lines 176-178.

Comment 13

Education and training focused on the importance of proper specimen collection, timing of collection, and testing could help avoid possible false negative results and SARS-CoV-2 transmissions

The authors should declare this training and method in the early of manuscript.

This has been added to the study design section: see lines 112-113.

Comment 14

Line 187: Another main limitation of this quality initiative is the absence of data available to assess the cost-benefits and impact of Lucira implementation in the testing settings. Cost analyses were not possible given Lucira kits and NAAT reagents and consumables for clinical testing were provided by provincial and federal initiatives to support SARS-CoV-2 testing.

I think this issue did not relate to this topic.

Cost remains an important practical consideration when evaluating any laboratory test.

Comment 15:

The language in limitation part should be revised.

The language in this section has been reviewed.

September 29, 2023

Dr. Elizabeth Simms
Dalhousie University
Department of Medicine, Department of Pathology
5820 University Avenue
Dickson Building, 5th floor
Halifax, Nova Scotia B3H2Y9
Canada

Re: Spectrum02772-23R2 (Real-world evaluation of the Lucira Check-It COVID-19 loop-mediated amplification (LAMP) test)

Dear Dr. Elizabeth Simms:

Your manuscript has been accepted, and I am forwarding it to the ASM Journals Department for publication. You will be notified when your proofs are ready to be viewed.

Sincerely,

Tulip Jhaveri
Editor, Microbiology Spectrum
